# miRNA-27a is essential for bone remodeling by modulating p62-mediated osteoclast signaling

**Shumin Wang[1], Eri O Maruyama[2], John Martinez[1], Justin Lopes[2], Trunee Hsu[3], Wencheng Wu[1], Wei Hsu[1,2,4,5,6]\*, Takamitsu Maruyama[1,2]\***

[1]University of Rochester Medical Center, Rochester, United States; [2]The Forsyth Institute, Cambridge, United States; [3]Case Western Reserve University, Cleveland, United States; [4]Faculty of Medicine, Harvard University, Boston, United States; [5]Harvard School of Dental Medicine, Boston, United States; [6]Harvard Stem Cell Institute, Cambridge, United States

**Abstract** The ability to simultaneously modulate a set of genes for lineage-specific development has made miRNA an ideal master regulator for organogenesis. However, most miRNA deletions do not exhibit obvious phenotypic defects possibly due to functional redundancy. miRNAs are known to regulate skeletal lineages as the loss of their maturation enzyme Dicer impairs bone remodeling processes. Therefore, it is important to identify specific miRNA essential for bone homeostasis. We report the loss of MIR27a causing severe osteoporosis in mice. MIR27a affects osteoclast-mediated bone resorption but not osteoblast-mediated bone formation during skeletal remodeling. Gene profiling and bioinformatics further identify the specific targets of MIR27a in osteoclast cells. MIR27a exerts its effects on osteoclast differentiation through modulation of Squstm1/p62 whose mutations have been linked to Paget's disease of bone. Our findings reveal a new MIR27a-p62 axis necessary and sufficient to mediate osteoclast differentiation and highlight a therapeutic implication for osteoporosis.

\*For correspondence:
wei.hsu@hsdm.harvard.edu
(WH);
tmaruyama@forsyth.org (TM)

**Competing interest:** The authors declare that no competing interests exist.

## Editor's evaluation

The study has been constructively revised with compelling evidence. The addition of pit resorption and actin formation assay along with the status of expression of osteoclast fusion markers have made the article more systematic making the findings fundamentally impactful. The study is novel and contributes significantly to bone biology research.

## Introduction

miRNA is a small non-coding RNA, base-pairing with complementary sequences of mRNA to control gene expression at post-transcriptional and translational levels (*Bartel, 2004*; *Baek et al., 2008*). In animals, miRNA recognizes the 3' untranslated region of their targets via a small stretch of seed sequences. A single miRNA may simultaneously affect the expression of hundreds of genes (*Hausser and Zavolan, 2014*; *Stark et al., 2005*). Because of its potential to modulate the same biological process at various steps, miRNA has been postulated to function as a master regulator for organogenesis (*Ebert and Sharp, 2012*; *Li and Rana, 2014*), similar to the transcription factor capable of turning on a set of genes for lineage-specific development. However, possibly due to functional redundancy, most of the miRNA deletions do not cause phenotypic alterations (*Miska et al., 2007*; *Park et al.,*

*2012*). With only limited evidence (*Croce, 2009*; *Shenoy and Blelloch, 2014*), it has been difficult to prove this concept.

The skeleton is constantly remodeled to maintain a healthy structure after its formation (*Manolagas and Jilka, 1995*). This lifelong process is called bone remodeling in which old bone is removed from the skeleton, followed by replaced with new bone. Therefore, a balance of osteoclast-mediated bone resorption and osteoblast-mediated bone formation is essential for bone metabolism (*Manolagas and Jilka, 1995*). Dysregulation of bone remodeling causes metabolic disorders, e.g., osteopetrosis, osteoporosis, and Paget's disease (*Feng and McDonald, 2011*). The current treatments have major limitations leading to the exploration of new therapeutic strategies. Studies of Dicer, an RNase III endonuclease involved in the maturation of miRNAs, suggest their importance in the development of the skeletal lineages during bone remodeling (*Gaur et al., 2010*; *Mizoguchi et al., 2010*). However, the specific miRNA(s) required for the differentiation of osteoclasts and osteoblasts remains largely unclear. The *Mir23a* cluster consists of *Mir23a*, *Mir27a*, and *Mir24-2*. Aberrant regulation of *Mir23a* and *Mir27a* has been associated with osteoporotic patients and increased bone fracture risks (*Seeliger et al., 2014*; *Guo et al., 2016*; *You et al., 2016*). The effect of MIR23a and MIR27a on the differentiation of osteoblast or osteoclast cells has been shown by in vitro overexpression studies (*Guo et al., 2016*; *Hassan et al., 2010*; *Guo et al., 2018*). Interference of MIR23a or MIR27a by the use of inhibitor/sponge has implied their role in osteoblast and osteoclast cells (*Hassan et al., 2010*; *Zeng et al., 2017*). However, due to the cross-reactivity of the RNA inhibitor among family members that share common seed motifs (*Androsavich et al., 2016*; *Ebert et al., 2007*), the inhibitor assay may not truthfully reflect the function of the target miRNA. Therefore, the genetic loss-of-function study remains the most rigorous method for determining the endogenous function of MIR23a and MIR27a as well as testing the removal of *Mir23a~27*a sufficient to cause bone loss.

We have performed mouse genetic analyses to definitively assess the requirement of MIR23a and MIR27a for skeletogenesis and homeostasis. Surprisingly, the skeletal phenotypes developed in newly established loss-of-function mouse models reveal findings different from previous reports based on the gain-of-function analyses. Severe loss of bone mass developed in mice with the deletion of *Mir23a~27*a or *Mir27a*. MIR23a~27a is dispensable for osteoblast-mediated bone formation. However, compelling evidence supports that MIR27a is essential for osteoclastogenesis and osteoclast-mediated bone resorption during bone remodeling. Gene expression profiling and bioinformatics analyses further identify osteoclast-specific targets of MIR27a. We demonstrated that MIR27a exerts its effects on osteoclast differentiation through modulation of a new and essential target Squstm1/p62. MIR27a is necessary and sufficient to mediate osteoclast differentiation and as a biomarker and therapeutic target for osteoporosis.

## Results

### The loss of MIR23a~27a in mice causes low bone mass phenotypes

To determine the requirement of MIR23a~27a for skeletal development and maintenance, we created a new mouse model with the deletion of *Mir27a* and *Mir23a*. The CRISPR/Cas9 gene edition method was used to establish the mouse strain carrying the *ΔMir23a~27*a allele (*Figure 1A*). PCR analysis of the *Mir23a* cluster demonstrated the sgRNA(single guide RNA)-mediated deletion and the reduction of 500 bp in the wild-type to 303 bp in the mutants (*Figure 1B*). Sequencing analysis also confirmed the expected genome editing (data not shown). Next, we tested if the deletions affect the expression of other miRNA molecules, generated from the same RNA precursor, within the same cluster. Semi-quantitative RT-PCR (qRT-PCR) analysis revealed that only MIR23a~27a are disrupted in the *ΔMir23a~27*a mutants (*Figure 1C*). These results indicated our success in establishing mouse models deficient for *Mir23a~27*a.

Mice heterozygous for *ΔMir23a~27*a are viable and fertile. Intercross between the heterozygotes successfully obtained the homozygous mutants without any noticeable skeletal deformity, suggesting that MIR23a~27a are not required for the developmental processes (*Figure 1—figure supplement 1*). Next, we examined if their deletions affect the homeostatic maintenance of the bone in adults. At 3 months, von Kossa staining and three-dimensional (3D) μCT analyses of the *ΔMir23a~27*a femurs revealed significant loss of the trabecular bone volume in both sexes (*Figure 1D–G*; BV/TV, n=3, mean ± SD; student t-test). Much more severe osteoporotic defects were detected in the 7-month-old

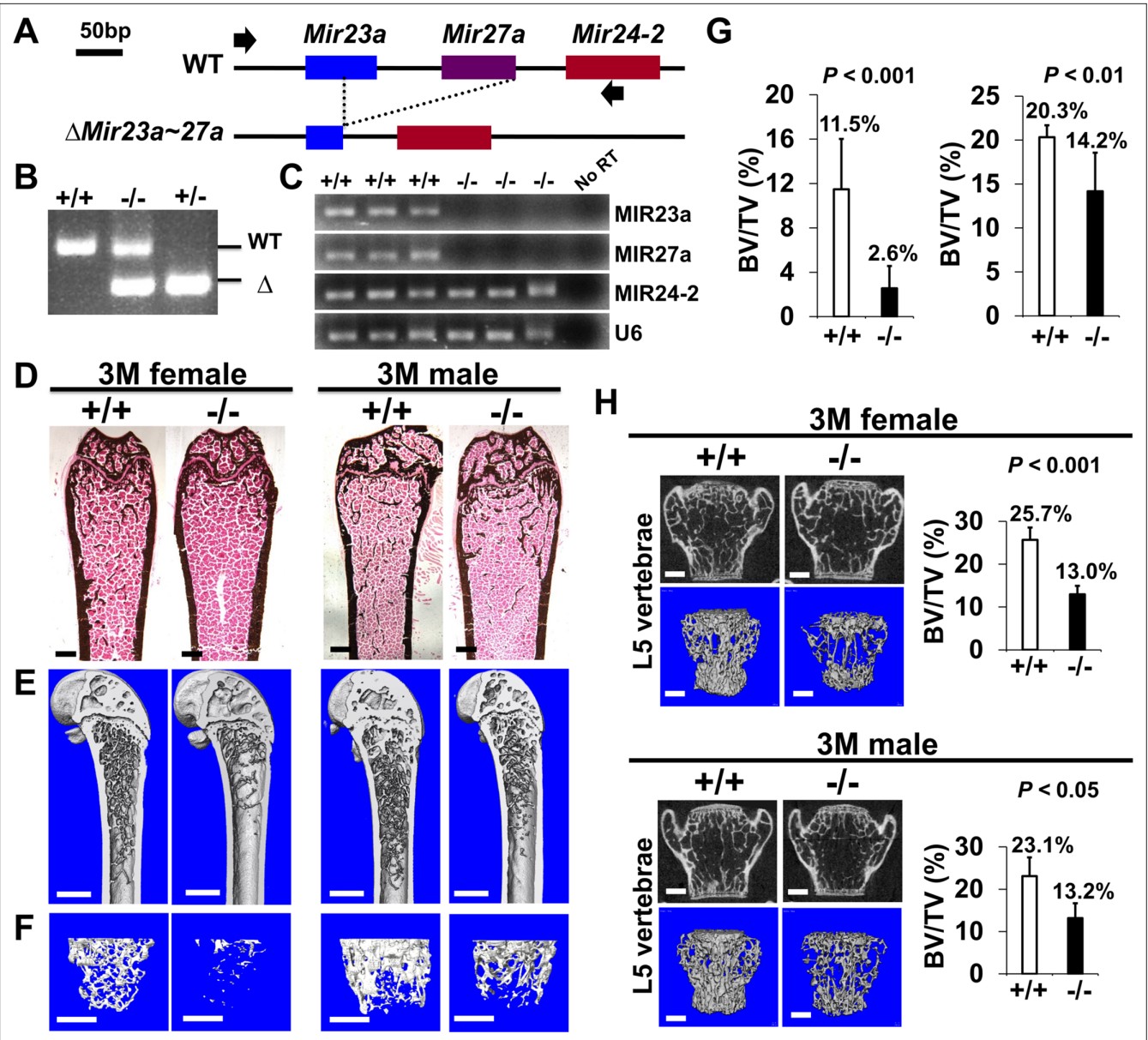

**Figure 1.** Low bone mass phenotypes in mice deficient for *Mir23a~27*a. (**A**) Diagrams illustrate the *Mir23a* cluster, consisting of *Mir23a*, *Mir27a*, and *Mir24-2* (WT), and the creation of mouse strains deficient for *Mir23a~27*a (*ΔMir23a~27*a) by CRISPR/Cas9 genome editing. Broken lines and arrows indicate the deleted genomic regions and primers used for PCR genotyping analysis, respectively. (**B**) PCR analysis examines the *Mir23a* cluster for genotyping the wild-type (+/+), heterozygous (+/−), and homozygous (−/−) for miR-23a~27 a mice. The mutant (Δ) alleles with deletion of *Mir23a~27*a result in the generation of shorter PCR products. (**C**) RT-PCR analysis of the MIR23a, MIR27a, and MIR24-2 RNAs reveals the disruption of specific miRNA(s) in the mutants. The analysis of small non-coding RNA U6 is used as an internal control. Femurs of the 3-month-old (3 M) wild-type (+/+) and mutant (−/−) males and females were analyzed by μCT scanning (**E–F**), followed by sectioning and von Kossa staining (**D**). Reconstructed μCT images of the distal femur (**E**) and femoral metaphysis (**F**) were subject to quantitative analysis for trabecular bone volume (**G**). (**H**) Spines of the 3-month-old (3 M) *ΔMir23a~27*a males and females were analyzed by μCT scanning. Images show the μCT scanned wild-type (+/+) and mutant (−/−) L5 vertebrae (top) and three dimensional (3D)-rendered trabecular bone (bottom). Quantitative analyses of trabecular bone volume per total volume in the femurs (BV/TV(bone volume to total volume fraction), n=7, mean ± SD; student t-test) and vertebrates (BV/TV, n=4 for female and n=3 for male, mean ± SD; student t-test) are shown in graphs. Images (**D–F and H**) are representatives of three independent experiments. Scale bars, 500 μm (**D–F and H**).

The online version of this article includes the following source data and figure supplement(s) for figure 1:

**Source data 1.** Statistical data for *Figure 1G–H*.

**Source data 2.** Raw gel image of *Figure 1B*.

**Source data 3.** Uncropped gel image of *Figure 1B* with labels.

*Figure 1 continued on next page*

*Figure 1 continued*

**Source data 4.** Raw gel image of *Figure 1C* -1.

**Source data 5.** Raw gel image of *Figure 1C* -2.

**Source data 6.** Uncropped gel image of *Figure 1C* with labels-1.

**Source data 7.** Uncropped gel image of *Figure 1C* with labels-2.

**Source data 8.** Uncropped gel image of *Figure 1C* with labels-3.

**Source data 9.** Uncropped gel image of *Figure 1C* with labels-4.

**Figure supplement 1.** MIR23a and MIR27a are dispensable during skeletal development.

**Figure supplement 1—source data 1.** Statistical data for *Figure 1—figure supplement 1*.

**Figure supplement 2.** The loss of MIR23a~27a causes an osteopenic phenotype in mice.

**Figure supplement 2—source data 1.** Statistical data for *Figure 1—figure supplement 2D*.

**Figure supplement 3.** The loss of MIR23a~27a does not affect cortical bone thickness.

**Figure supplement 3—source data 1.** Statistical data for *Figure 1—figure supplement 3*.

mutant females of *ΔMir23a~27*a (*Figure 1—figure supplement 2*; BV/TV, n=3, mean + SD; student t-test). However, cortical bone thickness does not seem to be affected by the deletion of *Mir23a~27*a (*Figure 1—figure supplement 3*). Next, we examined if similar bone loss phenotypes can be detected in the vertebrae where age-related changes in the trabecular architecture are minimal. Therefore, we examined the vertebrae of *ΔMir23a~27*a and identified drastic reductions in vertebral bone mass associated with the mutations (*Figure 1H*). These data demonstrated that MIR23a~27a is required for homeostatic maintenance of the bone.

## Osteoblast-mediated bone formation is not affected by the loss of MIR23a~27a

Proper maintenance of the skeleton requires balanced bone formation and resorption during bone remodeling. The bone loss phenotypes caused by the deletion of *Mir23a~27*a are likely to be associated with an imbalanced bone formation and resorption mediated by osteoblasts and osteoclasts, respectively. Therefore, we examined if bone formation and osteoblast activities are affected by the loss of MIR23a~27a. New bone formation was analyzed by double labeling with alizarin red and calcein at 3 months. Quantitative analyses did not reveal a significant difference in bone formation rate per unit of bone surface (BFR/BS) caused by the mutation (*Figure 2—figure supplement 1A*). In addition, the numbers of osteoblast cells positive for type 1 collagen (Col1) and Osteopontin (OPN) lining the trabecular BS remain comparable between the wild-type and homozygous littermates (*Figure 2—figure supplement 1B*), indicating that osteoblast-mediated bone formation is not affected by the *Mir23a~27*a deletion. The results suggested that MIR27a and MIR23a are not required for osteoblastogenesis and osteoblasts-mediated bone formation.

## MIR23a~27a regulates osteoclast differentiation

To determine if the loss of MIR23~27a affects bone resorption, we first examined osteoclast number by tartrate-resistant acid phosphatase (TRAP) staining. An increase of TRAP+ osteoclasts was detected in the 3-month-old *ΔMir23a~27* a males and females (*Figure 2*). When the number of TARP+ osteoclast cells in the total bone area (N.Oc/T.Ar), the ratio of TRAP + BS (Oc.S/BS), and the number of TRAP +osteoclast cells lining the BS (N.Oc/BS) were measured, we found that these parameters associated with bone resorption are significantly elevated in the mutants (*Figure 2*; n=5, mean ± SD; student t-test). In addition, there is a ~threefold increase in the number of Cathepsin K-expressing osteoclast cells lining the trabecular BS (*Figure 2*; n=5, mean ± SD; student t-test). These results support that the loss of MIR23a~27a stimulates osteoclastogenesis, leading to an elevation of bone resorption.

Next, to determine the role of MIR23a~27a in osteoclastogenesis, we analyzed cell populations associated with the differentiation of the osteoclast cells. During hematopoiesis, a common myeloid progenitor gives rise to monocytes that are precursors of several cell types, including dendritic cells, macrophages, and osteoclasts (*Charles and Nakamura, 2014*). Osteoclast precursors are known to derive from a monocyte population positive for CD11b and negative for Gr-1 (*Yao et al., 2006*).

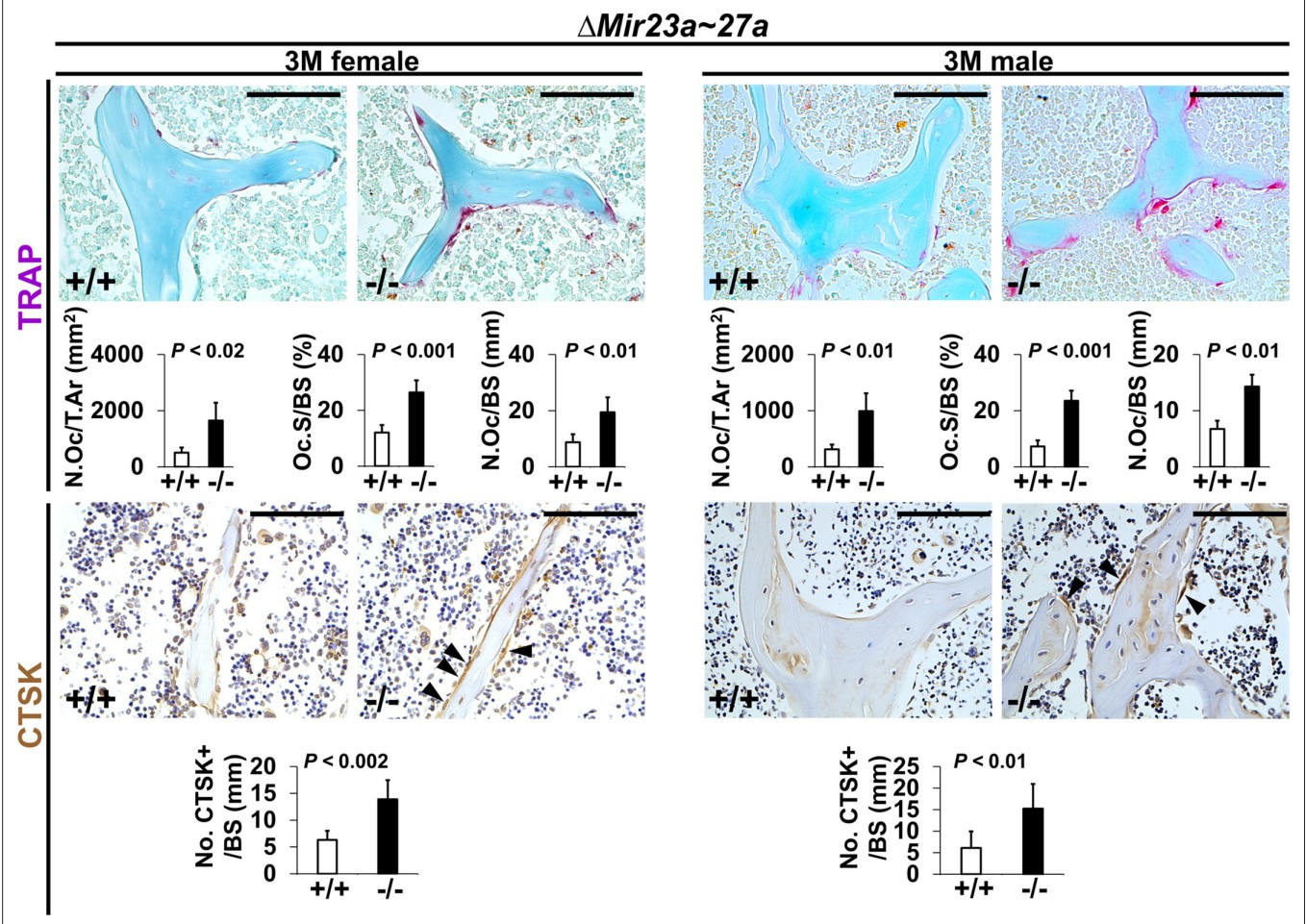

**Figure 2.** Increased number of osteoclast cells in the *ΔMir23a~27*a mice. Sections of the 3-month-old (3 M) *ΔMir23a~27*a males and females were analyzed by tartrate-resistant acid phosphatase (TRAP) staining and immunostaining of Cathepsin K (CTSK). Graphs show quantitative analyses of positively stained cells in the wild-type (+/+) and mutant (−/−) distal femurs (No. of cell+/BS, n=5, mean ± SD; student t-test). Histomorphometric parameters of bone resorption are evaluated by the number of osteoclast/bone area (N.OC/T.Ar), osteoclast surface/bone surface (OC.S/BS), and osteoclast number/bone surface (N.OC/BS, n=5, mean ± SD; student t-test). Images are representatives of five independent experiments. Scale bars, 100 μm.

The online version of this article includes the following source data and figure supplement(s) for figure 2:

**Source data 1.** Statistical data for *Figure 2*.

**Figure supplement 1.** Bone formation and osteoblastogenesis are not affected by the *ΔMir23a~27*a deletion.

**Figure supplement 1—source data 1.** Statistical data for *Figure 2—figure supplement 1*.

Therefore, we examined CD11b+ and Gr-1– monocytes to see if the *Mir23a~27*a deletion affects the osteoclast precursor population. Fluorescence-activated cell sorting(FACS) analysis revealed that the CD11b+ and Gr-1– population was not affected in the *ΔMir23a~27*a bone marrow of both male and female mice (*Figure 3A and B*). The CD11b+/CD11c+ dendritic cell population, derived from the monocytes, was also unaffected by the *Mir23a~27*a deletion (*Figure 3A and B*). Although the precursors are not affected, MIR23a~27a may play a role in osteoclast differentiation.

To test osteoclast differentiation affected by the loss of MIR23a~27a, an ex vivo analysis was performed with cultures of cells seeding at two different densities. Cells isolated from the bone marrow were cultured in the presence of M-CSF to obtain bone marrow-derived macrophages, followed by differentiation into osteoclasts with the treatment of RANKL. TRAP staining was then used to assess the extent of osteoclast differentiation. The number of TRAP+ cells was significantly increased by the loss of MIR23a~27a (*Figure 3C*; n=5, mean ± SD; student t-test).

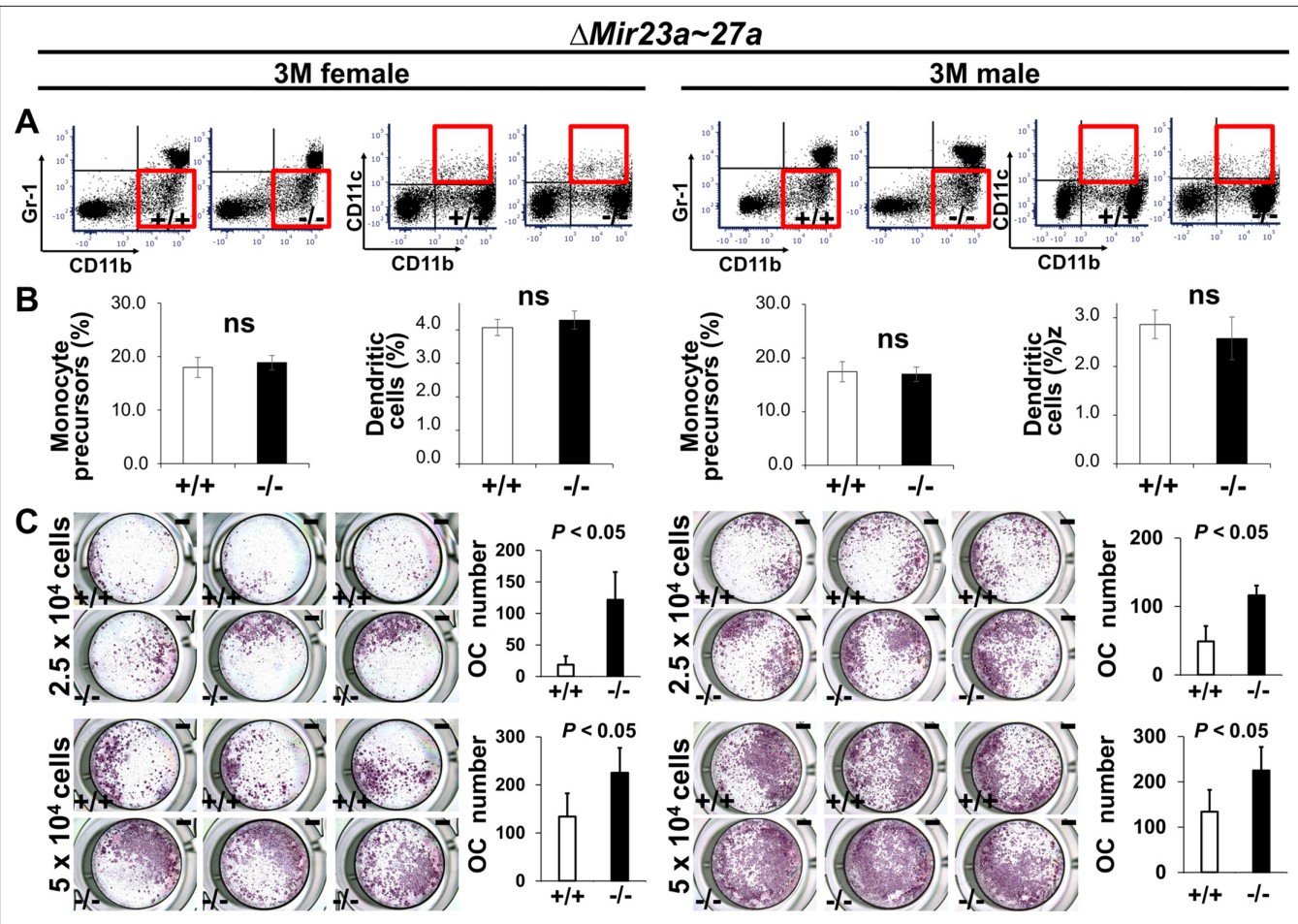

**Figure 3.** MIR23a~27a regulates osteoclast (OC) differentiation. (**A**) FACS analysis examines the CD11b+/Gr-1− and CD11b+/CD11c+ populations for monocyte precursors and dendritic cells, respectively. Images are representatives of three independent experiments. (**B**) The bone marrow of 3-month-old (3 M) wild-type (+/+) and $\Delta Mir23a\sim27$a (−/−) males and females show no significant difference (ns; n=3, mean ± SD; student t-test). (**C**) Cells isolated from the bone marrow were induced for OC differentiation with Receptor activator of nuclear factor kappa- B ligand(RANLK) and macrophage colony-stimulating factor(MCSF) for 5 days (top, 2.5×10⁴ cells/well and bottom, 5×10⁴ cells/well). The number of OC cells positive for tartrate-resistant acid phosphatase (TRAP) staining is significantly enhanced in the mutant cultures (n=3, mean ± SD; student t-test). Scale bars, 1 mm (**C**).

The online version of this article includes the following source data for figure 3:

**Source data 1.** Statistical data for *Figure 3*.

## MIR27a is an essential regulator for osteoclast-mediated skeletal remodeling

Using the miRPath Reverse-Search module, we searched and ranked miRNAs whose targets are accumulated in osteoclast differentiation-related genes based on the enrichment of the targets in the Kyoto Encyclopedia of Genes ad Genomics (KEGG: mmu04380; *Kanehisa and Goto, 2000*; *Kanehisa et al., 2008*). Among them, MIR27a was predicted as the top three candidates to regulate osteoclast differentiation (*Figure 4—figure supplement 1*, p<2.0×10⁻⁷¹). Its sister gene MIR27b contains the same seed sequences also ranked in the top three. However, MIR23a had a lower estimated rank, suggesting that MIR27a alone may be sufficient to exert osteoclast regulation (*Figure 4—figure supplement 1*, p<4.9×10⁻²⁹). To test this hypothesis, we created another mouse strain with the deletion of only MIR27a using CRISPR/Cas9 genome editing (*Figure 4A*). PCR analysis revealed the sgRNA-mediated deletion causes the reduction of 500 bp in the wild-type to 474 bp in the mutants (*Figure 4B*). Sequencing of the PCR products confirmed the genomic deletion (data not shown). Next, semi-qRT-PCR analysis indicated that only MIR27a are disrupted in the $\Delta Mir27a$ mutants, suggesting the deletions does not affect the expression of other miRNA generated from the same RNA precursor

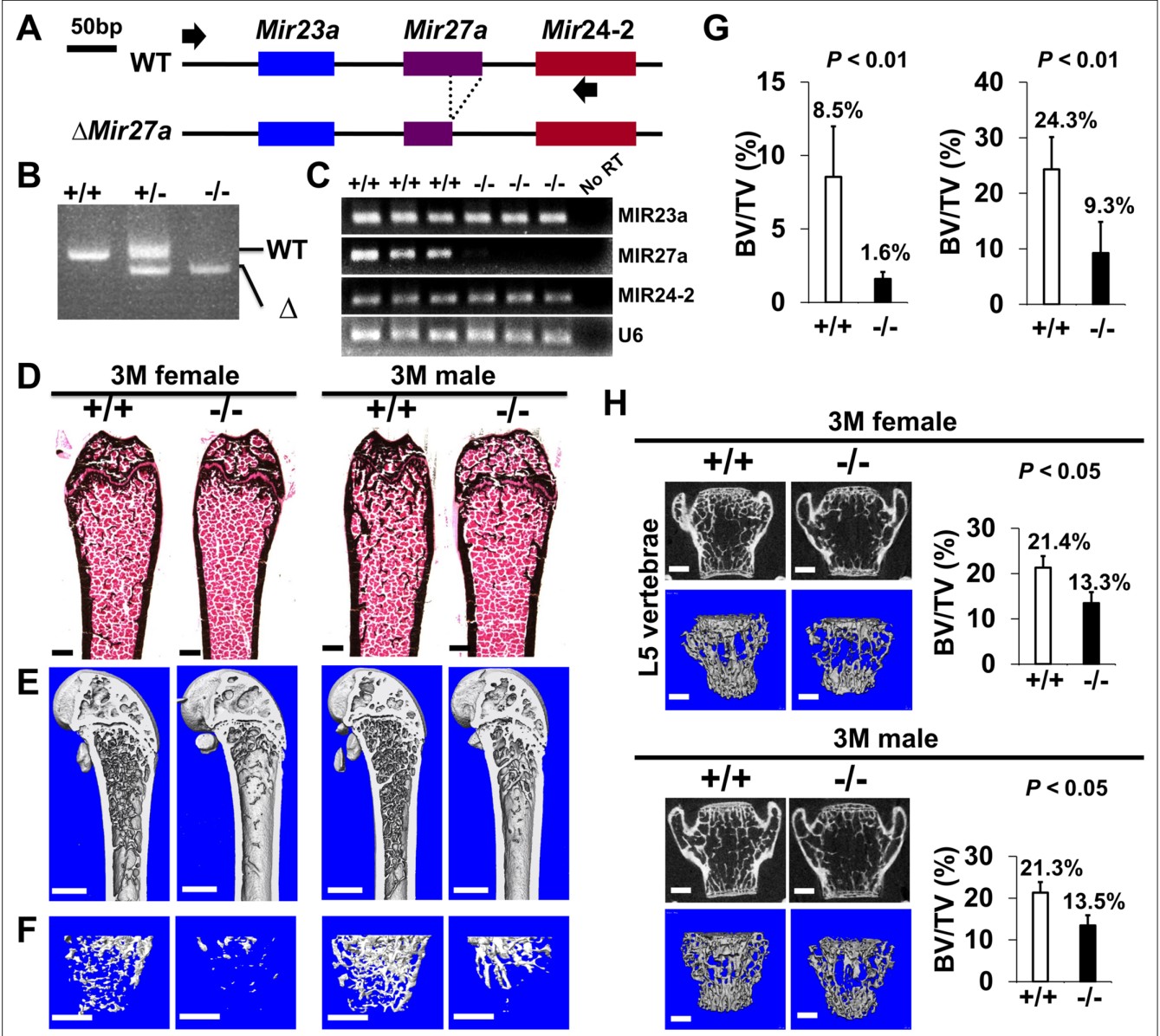

**Figure 4.** The loss of MIR27a alone causes osteoporotic defects. (**A**) Diagrams illustrate the *Mir23a* cluster (WT), and the creation of mouse strains deficient for *Mir27a* (*ΔMir27a*) by CRISPR/Cas9 genome editing. Broken lines and arrows indicate the deleted genomic regions and primers used for PCR genotyping analysis, respectively. (**B**) PCR-based genotyping identifies the wild-type (+/+), heterozygous (+/–), and homozygous (–/–) for miR-27a mice showing the mutant (Δ) alleles with deletion of *Mir27a* result in the generation of shorter PCR products. (**C**) RT-PCR analysis reveals the disruption of MIR27a but not MIR23a and MIR24-2 RNAs in *ΔMir27a* mutants. The analysis of small non-coding RNA U6 is used as an internal control. Femurs of the 3-month-old (3 M) wild-type (+/+) and mutant (–/–) males and females were analyzed by μCT scanning (**E–F**), followed by sectioning and von Kossa staining (**D**). The three dimensional (3D) rendered μCT images of the distal femur (**E**) and femoral metaphysis (**F**) were subject to quantitative analysis for trabecular bone volume (**G**). (**H**) Spines of the 3-month-old (3 M) *ΔMir27a* males and females were analyzed by μCT scanning. Images show the μCT scanned wild-type (+/+) and mutant (–/–) L5 vertebrae (top) and 3D-rendered trabecular bone (bottom). Quantitative analyses of trabecular bone volume per total volume in the femurs (BV/TV, n=5 for female and n=4 for male, mean ± SD; student t-test) and vertebrates (BV/TV, n=3, mean ± SD; student t-test) are shown in graphs. Images (**D–F and H**) are representatives of three independent experiments. Scale bars, 500 μm (**D–F and H**).

The online version of this article includes the following source data and figure supplement(s) for figure 4:

**Source data 1.** Statistical data for *Figure 4G–H*.

**Source data 2.** Raw gel image of *Figure 4B*.

**Source data 3.** Uncropped gel image of *Figure 4B* with labels.

**Source data 4.** Raw gel image of *Figure 4C*.

*Figure 4 continued on next page*

*Figure 4 continued*

**Source data 5.** Uncropped gel image of *Figure 4C* with labels-1.

**Source data 6.** Uncropped gel image of *Figure 4C* with labels-2.

**Source data 7.** Uncropped gel image of *Figure 4C* with labels-3.

**Source data 8.** Uncropped gel image of *Figure 4C* with labels-4.

**Figure supplement 1.** Identification of miRNA candidates in osteoclast differentiation pathway.

**Figure supplement 2.** The loss of MIR27a does not affect cortical bone thickness.

**Figure supplement 2—source data 1.** Statistical data for *Figure 4—figure supplement 1*.

(*Figure 4C*). These results demonstrated our success in establishing mouse models deficient for MIR27a.

Mice heterozygous and homozygous for *ΔMir27a* are viable and fertile similar to the *Mir23a~27a* deletion. As anticipated, there was no noticeable skeletal deformity associated with the loss of MIR27a, suggesting its dispensable role in the developmental processes. However, von Kossa staining and 3D µCT analyses of the 3-month-old male and female femurs of *ΔMir27a* revealed significant loss of the trabecular bone volume (*Figure 4D–G*; BV/TV, n=3, mean ± SD; student t-test), while cortical bone thickness was not affected (*Figure 4—figure supplement 2*). Drastic bone loss phenotypes can also be detected in the *ΔMir27a* vertebrae where age-related changes in the trabecular architecture are minimal (*Figure 4H*). These data demonstrated the essential role of MIR27a in skeletal remodeling.

Using double labeling with alizarin red and calcein for quantitative analyses, we did not reveal a significant difference in BFR/BS (*Figure 5—figure supplement 1A*), and Col1+ and OPN+ osteoblast cells lining the trabecular BS (*Figure 5—figure supplement 1B*) at 3 months, suggesting that osteoblast-mediated bone formation is not affected by the *Mir27a* deletion. However, we detected a significant increase of TRAP+ and Cathepsin K-expressing osteoclast cells lining the trabecular BS in the 3-month-old *ΔMir27a* males and females (*Figure 5A* and *Figure 5—figure supplement 2*; n=5, mean ± SD; student t-test), supporting that the loss of MIR27a stimulates osteoclastogenesis, leading to elevated bone resorption. While the CD11b+/Gr-1– and CD11b+/CD11c+ osteoclast precursor populations were not affected by the *Mir27a* deletion (*Figure 5—figure supplement 3A*, B), osteoclast differentiation was significantly increased by the loss of MIR27a (*Figure 5B*; n=5, mean ± SD; student t-test). Furthermore, the loss of a single MIR27a recapitulates the osteoporotic phenotypes caused by the double deletion of *Mir23a* and *Mir27a*, suggesting that MIR27a is responsible for skeletal maintenance through the modulation of bone remodeling processes. The results suggested that MIR27a functions as a negative regulator in osteoclast differentiation. To test this possibility, we overexpressed MIR27a in cells undergoing osteoclast differentiation. High levels of MIR27a significantly reduce the number of differentiated osteoclast cells (*Figure 5C*). Our findings demonstrated that MIR27a is necessary and sufficient to modulate osteoclast differentiation. Osteoclastogenesis mediated by MIR27a is essential for bone remodeling and homeostasis.

## MIR27a regulates osteoclast differentiation through the modulation of p62

To elucidate the mechanism underlying osteoclast differentiation regulated by MIR27a, we first used a bioinformatics approach to identify its potential targets (*Figure 6A*). The TarBase computationally predicted 2312 target genes for MIR27a (*Vlachos et al., 2015a*). Furthermore, there were 154 genes associated with osteoclast differentiation based on the Kyoto Encyclopedia of Genes ad Genomics (KEGG: mmu04380; *Kanehisa and Goto, 2000*; *Kanehisa et al., 2008*). The miRPath software further identified 26 targets overlapping with the osteoclast-related genes (*Vlachos et al., 2015b*). Next, we examined the transcript level of these 26 targets in wild-type and *ΔMir27a* osteoclast cells. qRT-PCR analyses revealed that five of these targets, *Snx10*, *Map2k7*, *Ctsk*, *Tgfbr1*, and *Sqstm1*, are significantly upregulated by the loss of MIR27a (*Figure 6B*, $p < 0.05$, n=3; two-sided student t-test). To test if these genes were the direct targets of MIR27a, we performed 3'UTR-reporter assays. The expression of MIR27a significantly downregulated the luciferase activity associated with the 3'UTR of *Sqstm1*, *Tgfbr1*, *Snx10*, and *Map2k7* but not *Ctsk* (*Figure 6C*, *; $p < 0.01$, n=3, mean ± SD; two-sided student t-test). As Cathepsin K is a protease expressed in the mature osteoclast cells, its alteration at the

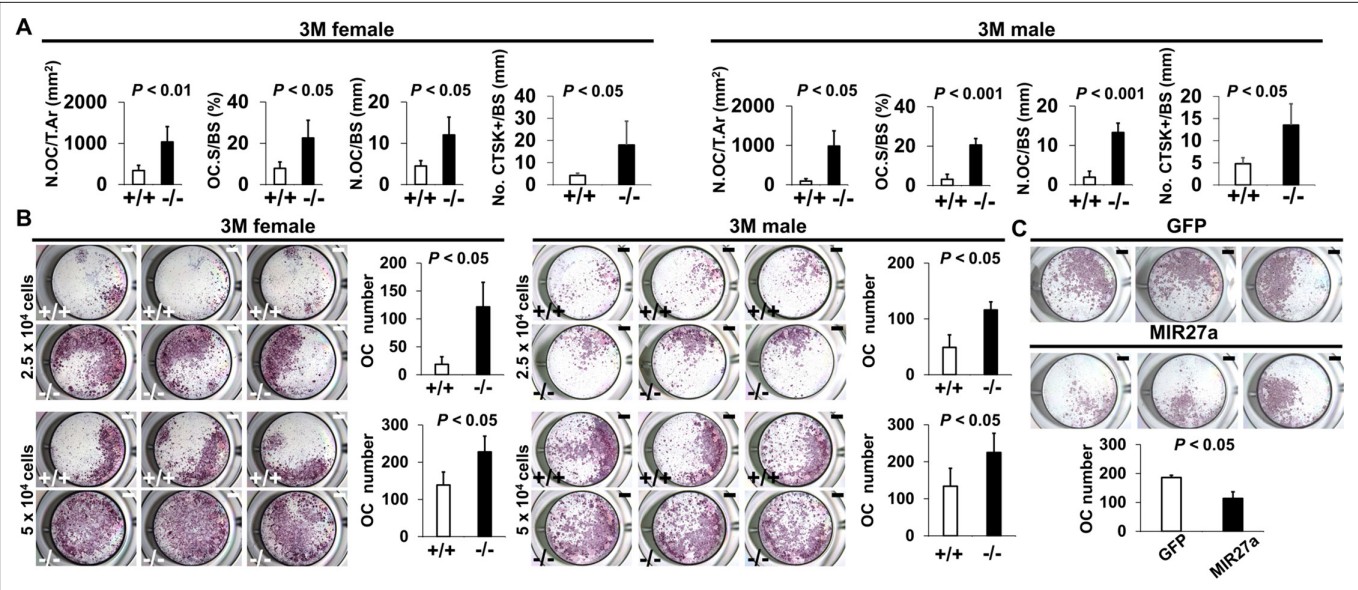

**Figure 5.** MIR27a is necessary and sufficient to repress osteoclast (OC) differentiation. (**A**) Graphs show quantitative analyses of tartrate-resistant acid phosphatase (TRAP)+ and CTST + cells in the wild-type (+/+) and mutant (−/−) distal femurs detected in the stained section of the 3-month-old (3 M) *ΔMir27a* males and females (n=5 for female and n=4 for male, mean ± SD; student t-test). Histomorphometric parameters of bone resorption are evaluated by the number of OC/bone area (N.OC/T.Ar), OC surface/bone surface (OC.S/BS), OC number/bone surface (N.OC/BS), and CTSK + cells/ bone surface (No. CTSK+/BS). (**B**) Cells isolated from the wild-type (+/+) and mutant (−/−) bone marrows were induced for OC differentiation with RANLK and MCSF for 5 days (top, 2.5×10$^4$ cells/well and bottom, 5×10$^4$ cells/well). Graphs indicate the number of TRAP+ OC cells (n=3, mean ± SD; student t-test). (**C**) Cells isolated from the bone marrow were seeded (5×10$^4$ cells/well) and induced for OC differentiation using RANLK and MCSF for 5 days with the lentivirus-mediated expression of green fluorescent protein(GFP) (control) or MIR27a. The TRAP+ OC cell number is significantly decreased in the MIR27 overexpression cultures (n=3, p<0.05, mean ± SD; student t-test). Scale bars, 1 mm (**B and C**).

The online version of this article includes the following source data and figure supplement(s) for figure 5:

**Source data 1.** Statistical data for *Figure 5*.

**Figure supplement 1.** Bone formation and osteoblastogenesis are not affected by the *ΔMir27a* deletion.

**Figure supplement 1—source data 1.** Statistical data for *Figure 5—figure supplement 1*.

**Figure supplement 2.** Enhanced osteoclastogenesis in the *ΔMir27a* mice.

**Figure supplement 3.** Osteoclast precursor populations are not affected by the loss of MIR27a.

**Figure supplement 3—source data 1.** Statistical data for *Figure 5—figure supplement 3*.

transcript level is likely ascribed to indirect effects of increased osteoclastogenesis in *ΔMir27a*. The data indicated *Sqstm1*, *Tgfbr1*, *Snx10*, and *Map2k7* as direct targets of MIR27a.

*Sqstm1* also known as p62 whose gain of function mutations were linked to Paget's disease of bone with disruption of bone renewal cycle causing weakening and deformity (*Rea et al., 2006*). The deletion of p62 in mice also impaired osteoclast differentiation (*Durán et al., 2004*). Therefore, we performed a functional study to test the importance of the MIR27a-p62 regulatory axis during osteoblastogenesis. Cells isolated from the bone marrow were induced for osteoclast differentiation and the number of osteoclast cells positive for TRAP staining was counted to determine the outcome of the differentiation. The enhanced osteoclast differentiation in the *ΔMir27a* culture was significantly alleviated by the shRNA-mediated knockdown of p62 (*Figure 6D*, p<0.05, n=3, mean ± SD; two-sided student t-test). To assess osteoclast function, we first performed phalloidin staining to ensure actin ring formation was not affected by *Mir27a* deletion and p62 knockdown in OC cells (*Figure 6—figure supplement 1*). Bone resorption pit assay then revealed enhanced bone resorption activity of *ΔMir27a* osteoclasts that can be alleviated by p62 knockdown (*Figure 6E*, p<0.05, n=3, mean ± SD; two-sided student t-test). Next, qRT-PCR of osteoclast markers was performed to decipher the regulatory process of MIR27a-mediated osteoclast differentiation. The deletion of *Mir27a* elevated the expression of Rank, but this elevation could be alleviated by p62 knockdown (*Figure 6F*, p<0.05, n=3, mean ± SD; two-sided student t-test), indicating the importance of the MIR27a-p62 axis for

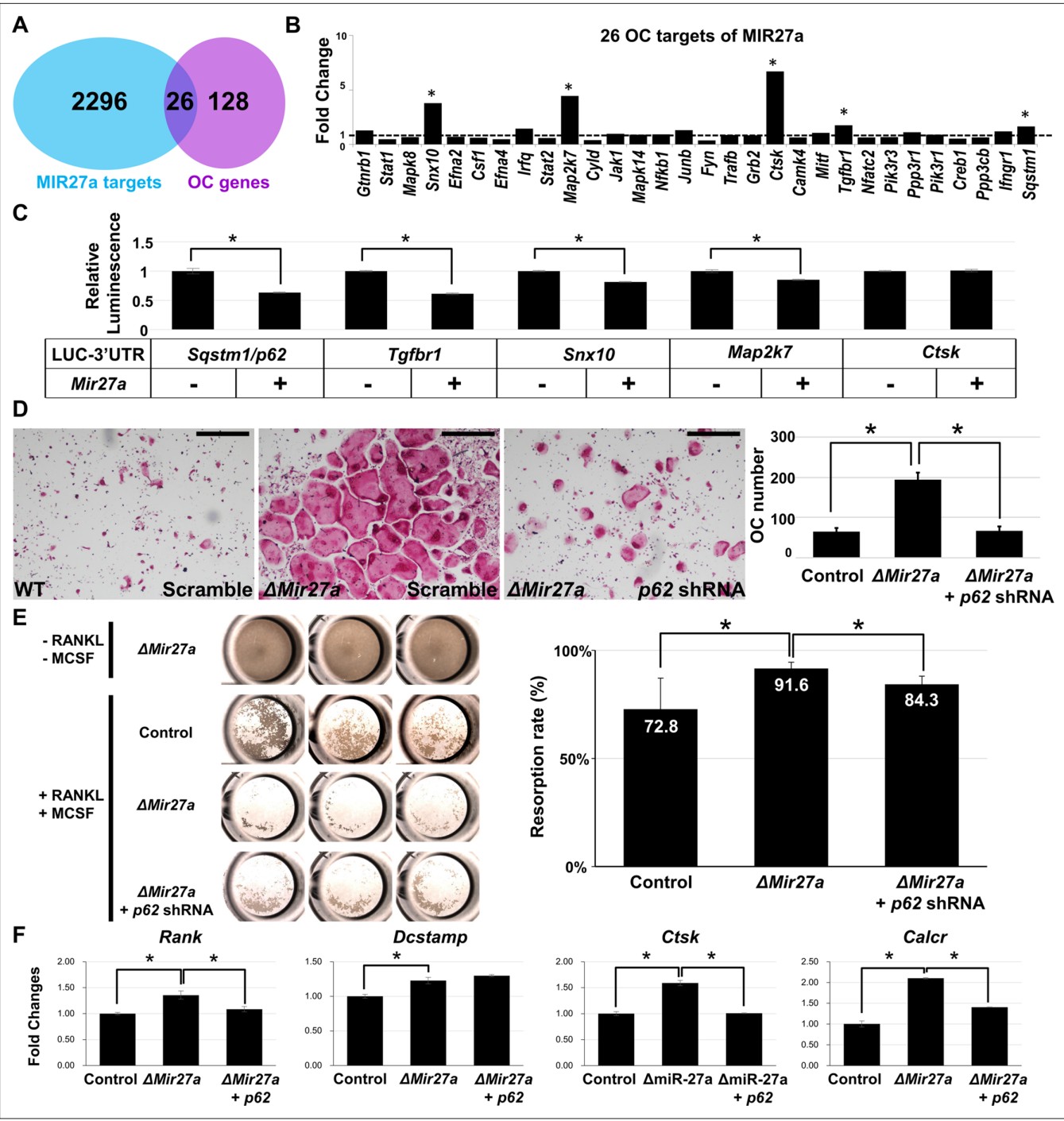

**Figure 6.** MIR27a-dependent regulation of osteoclast (OC) differentiation is mediated through p62 modulation. (**A**) Venn diagrams illustrate our strategy to identify the OC differentiation-associated genes (KEGG: mmu04380) that are directly regulated by MIR27a. (**B**) The 26 potential targets are examined by quantitative RT-PCR (qRT-PCR) analysis to detect the change of transcript levels in wild-type and *ΔMir27a* OC cells (n=3; *, p-value<0.05, two-sided student's t-test). (**C**) The 3'UTR-reporter assay examines five potential genes that are direct targets of MIR27a (n=3; *, p-value <0.01, means ± SD, two-sided student's t-test). (**D**) A functional study of Sqstm1 also known as p62 reveals the enhancement of OC differentiation caused by the loss of miR-27a is alleviated by lentiviral-shRNA-mediated knockdown. Tartrate-resistant acid phosphatase (TRAP) staining examines the number of mature OC cells in the control, *ΔMir27a*, and *ΔMir27a* plus shRNA-mediated knockdown of p62 (*ΔMir27a+p62 shRNA*) cultures (n=3; *, p-value<0.05, means ± SD, two-sided student's t-test). (**E**) Pit assay examined OC cells-mediated bone resorption rate of control, *ΔMir27a*, *ΔMir27a* plus shRNA-mediated p62 knockdown 5 days after differentiation (n=3; *, p-value<0.05, mean ± SD, two-sided student t-test). OC progenitors isolated from *ΔMir27a* mice were cultured without RANKL and MCSF as negative control (Top raw). (**F**) qRT-PCR examined the expression of OC markers after 3 days (*Rank* and *Dcstamp*)

*Figure 6 continued on next page*

*Figure 6 continued*

or 7 days (*Ctsk* and *Calcr*) culture of the control, *ΔMir27a*, and *ΔMir27a* plus shRNA-mediated knockdown of p62 OC cells (n=3; *, p-value<0.05, mean ± SD, student t-test). Scale bars, 500 μm.

The online version of this article includes the following source data and figure supplement(s) for figure 6:

**Source data 1.** Statistical data for *Figure 6*.

**Source data 2.** MIR27a targets.

**Figure supplement 1.** No effect of actin ring formation by the loss of MIR27a in osteoclast (OC) cells.

RANKL signaling. The expression of *Dcstamp*, essential for cell-cell fusion during osteoclastogenesis, was increased in *ΔMir27a* osteoclasts (*Figure 6F*, p<0.05, n=3, mean ± SD; two-sided student t-test). However, the increased level of *Dcstamp* in *ΔMir27a* osteoclasts could not be affected by p62 knockdown, implying that MIR27a-mediated regulation of osteoclast fusion is independent of p62. Mature osteoclast markers, e.g., Ctsk and Calcr, were significantly upregulated in *ΔMir27a* osteoclasts but reduced by p62 knockdown (*Figure 6F*, p<0.05, n=3, mean ± SD; two-sided student t-test). The results demonstrated that MIR27a-dependent osteoclast differentiation is mediated through the regulation of p62. The MIR27a-p62 regulatory axis plays an important role in osteoclastogenesis during bone remodeling.

## Discussion

The dysregulation of miRNA has been implicated in osteoporosis in menopausal women. Among 851 miRNAs tested MIR27a is one of the most significant genes downregulated in postmenopausal osteoporosis patients (*You et al., 2016*). However, it's not clear whether the alteration of miRNAs is the cause or consequence of the disease. Our genetic study presented here has demonstrated the loss of MIR23a~27a or MIR27a results in significant bone loss. The findings suggest a single miRNA deficiency can lead to severe osteoporotic defects, indicating an essential role of MIR27a in bone remodeling. Because osteoporosis is caused by an imbalance of osteoblast-mediated bone formation and osteoclast-mediated bone resorption, the conventional knockout model is ideal to decipher the regulatory processes underlying MIR27a-dependent pathogenesis. Our data also suggest that MIR27a is dispensable for osteoblast differentiation and bone formation as its deletion does not affect the number of osteoblast cells and BFRs. The results do not agree with the previous gain-of-function study indicating an inhibitory role of MIR27a in osteoblastogenesis (*Hassan et al., 2010*). Therefore, the association of osteoporosis with both upregulation and downregulation is possibly mediated through distinct mechanisms underlying the regulatory process of the *Mir23a* cluster.

This study provides compelling evidence to first demonstrate that MIR27a is essential for regulating the bone resorption process through modulation of osteoclast differentiation. The loss of MIR27a in mice leads to elevated numbers of osteoclast cells as well as increases in bone resorption activity. The inhibitory function of MIR27a on osteoclastogenesis is also in agreement with previous in vitro culture data showing its crucial downregulation among miRNAs associated with osteoclast differentiation (*Ma et al., 2016*). Furthermore, MIR23a upregulation has been detected in osteopetrosis patients (*Ou et al., 2014*). Although the number of osteoclast progenitors is comparable between the control and mutant, the deletion of *Mir27a* strongly accelerates the process of osteoclastogenesis. The isolated *ΔMir27a* exhibits a highly potent ability in differentiation and maturation, indicating cell-autonomous regulations of MIR27a in the osteoclast cell.

The gain-of-function mutations have linked p62 to the cause of Paget's disease of bone – a genetic disorder characterized by aberrant osteoclastic activity (*Rea et al., 2006*; *Laurin et al., 2002*). The knockout of p62 in mice further supports its critical role in osteoclastogenesis (*Rea et al., 2006*). Our identification and characterization of p62 as a direct downstream regulator of MIR27a established a new osteoclast signaling axis. Not only osteoclast differentiation and maturation are suppressed by high levels of p62 but also its reduction can alleviate excessive osteoclastogenesis caused by the loss of MIR27a. Our findings suggest the MIR27a-p62 regulatory axis is necessary and sufficient to regulate bone remodeling through the modulation of osteoclastogenesis. In addition, circulatory miRNAs were released in the cell-free form either bound with protein components or encapsulated with microvesicles. They are quite stable and found with variations in miRNA signature as biomarkers (*Garnero,*

*2008*). Therefore, the identification of MIR27a as essential for bone remodeling promises its use as a biomarker for early detection of bone destruction-associated diseases, risk prediction for bone fracture, as well as personalized treatment and monitoring of the treatment efficacy.

Hormone therapy is effective for the prevention and treatment of postmenopausal osteoporosis as estrogen reduction is a crucial pathogenic factor. Because of the well-documented side effects, e.g., cardiovascular events and breast cancer risk, estrogen-based therapies are now limited to short-term use (*Rossouw et al., 2002*; *Khosla and Hofbauer, 2017*). Another widely used treatment for osteoporosis is bisphosphonates, which possess a high affinity for bone minerals with inhibitory effects on osteoclast cells (*Khosla and Hofbauer, 2017*; *Russell et al., 2008*). However, there is a need for alternative treatments due to the side effects of bisphosphonates, e.g., atypical femur fracture and osteonecrosis of the jaw (*Russell et al., 2008*; *Shane et al., 2014*). In addition, the treatment effectiveness after 5 years remains uncertain (*Izano et al., 2020*). A better understanding of the bone remodeling process can help maximize anti-fracture efficacy and minimize adverse skeletal effects.

The clear demonstration of osteoporotic bone loss caused by the disruption of MIR27a suggests its supplementation be explored as a new therapeutic approach. The clinical application requires a system for osteoclast delivery of MIR27a. However, this may be complicated by the negative effects of the MIR23a cluster on osteoblast-mediated bone formation (*Zeng et al., 2017*). High levels of MIR23a in the mast cells also lead to bone loss through the release of extracellular vesicles by neoplastic mast cells (*Kim et al., 2021*). Therefore, targeting MIR27a to the bone resorption surfaces with synthetic compounds such as bisphosphonates or osteoclast-targeting molecules such as acid octapeptides with aspartic acid is critical for future clinical applications (*Dang et al., 2016*).

## Materials and methods

### Animals

The CRISPR/Cas9 gene edition strategy was used to generate *ΔMir23a~27*a and *ΔMir27a* mouse strains (*Cong et al., 2013*; *Mali et al., 2013*). Complementary oligonucleotides containing the *Mir23a* or *Mir27a* sgRNA target sequences were designed by CRISPR Design Tool (http://crispr.mit.edu) and inserted into the pX335 plasmid (Addgene, Cambridge, MA, USA), followed by DNA sequencing to verify the correct cloning. A mixture of sgRNA plasmid, Cas9 protein (NEB, Ipswich, MA, USA), and ssODN (Integrated DNA Technologies, Coralville, IA, USA) was injected into the pronuclei and cytoplasm of fertilized eggs (*Mashiko et al., 2013*; *Wu et al., 2016*). The survived embryos were transferred into the oviduct of pseudopregnant females for carrying to term. The founder lines were genotyped by PCR analysis using primers 5′-GAC CCA GCC TGG TCA AGA TA-3′ and 5′-GGA CTC CTG TTC CTG CTG AA-3′ to determine the success of the gene edition and germline transmission. Both male and female mice were used in this study. Care and use of experimental animals described in this work comply with the guidelines and policies of the University Committee on Animal Resources at the University of Rochester and IACUC at the Forsyth Institute.

### Genes

Total RNAs including miRNAs were isolated using a mirVana miRNA isolation kit (Thermo Fisher Scientific, Waltham, MA, USA), followed by polyadenylation using *Escherichia coli* Poly(A) polymerase (NEB), and reserve transcribed into DNA using Reverse Transcriptase (Thermo Fisher Scientific) and an anchor primer. Based on the dominant mature miRNA sequences, four to six nucleotides were added to the 5′ end to enhance the annealing specificity. To detect the expression of the miRNAs, the reverse transcription products were subject to PCR analysis using forward and reverse primers listed in *Supplementary file 1*. The PCR was performed by denaturation at 95°C for 2 min and 27 cycles of amplification (95°C for 20 s, 60°C for 10 s, and 70°C for 10 s). The Lentivirus-*Mir27a* (Cat #MmiR3347MR03, GeneCopoeia, Rockville, MD) and Lentivirus-shRNA *Sqstm1* (Product ID: MSH093992, Accession: NM_011018.3, GeneCopoeia) were used to express *Mir27a* and knockdown *Sqstm1*, respectively.

### Cells

Primary cells were harvested from bone marrows of the bilateral femur to obtain bone marrow cells. These cells were incubated with Mouse BD Fc Block (Cat #553142, BD Biosciences, San Jose, CA, USA) to reduce non-specific antibody staining caused by receptors for IgG, followed by FACS analysis

with anti-CD11b/Mac-1 (M1/70)-APC (17-0112-81, eBioscience/Thermo Fisher Scientific; 1:400), anti-granulocyte mAb: anti-Ly-6G/Gr-1 (RB6-8C5)-PE (12-5931-81, eBioscience; 1:10000), and anti-dendritic cell mAb: anti-CD11c (N418)-FITC (11-0114-81, eBioscience; 1:20) using LSR II (BD Biosciences). For osteoclast differentiation, isolated cells were cultured in αMEM containing 10% FBS, 1% L-glutamine, 1% non-essential amino acids, and 5 ng/ml M-CSF for 2 days, followed by the addition of RANKL 10 ng/ml (R&D, Minneapolis, MN, USA). The differentiated cells were then fixed with 10% formalin followed by TRAP staining (*Yao et al., 2006*). For bone resorption assay, $5 \times 10^4$ primary osteoclast progenitors isolated from 3-month-old control and *ΔMir27a* femur were seeded in 96 well Corning Osteo Assay Surface plate (Cat #3988, Corning, Glendale, AZ, USA) and cultured in 5% $CO_2$ with differentiation media at 37°C. After culturing for 5 days, the resorbed areas on the plates were imaged by a Nikon SMZ1500 microscope, followed by ImageJ analysis. For actin ring formation assay, $5 \times 10^4$ primary osteoclast progenitors isolated from 3-month-old control and *ΔMir27a* femur were seeded in 96 well plates. After 5 days of culture, the cells were fixed with 4% Paraformaldehyde(PFA) in PBS for 15 min, permeabilized with 0.1% Triton X-100 in PBS, and incubated with Phalloidin-Alexa 488 (Cat #A12379 Invitrogen) for 1 hr at room temperature. Nuclear counterstaining was performed by DAPI (4',6-diamidino-2-phenylindole).

## 3'UTR assay

The LUC-3'UTR reporter DNA plasmids contain the 3'UTR from *Sqstm1*, *Tgfbr1*, *Snx10*, *Map2k7*, and *Ctsk* fused to the end of a luciferase reporter gene (MmiT079315-MT06, MmiT096143-MT06, MmiT073076-MT06, MmiT099033-MT06, MmiT091679-MT06, and GeneCopoeia). C3H10T1/2 mesenchymal cells were transfected by the LUC-3'UTR without or with co-transfection of *Mir27a* (MmiR3347-MR04-50 and GeneCopoeia) using Lipofectamine 200 (Cat #11668027, Invitrogen, Waltham, MA, USA). The transcript stability and its translation efficiency in the presence or absence of *Mir27a* were determined by the luciferase assay 72 hr after the transfection using the Dual-Luciferase Reporter Assay System (Cat #1910, Promega, Madison, WI, USA). The luminescent intensity was measured by the SpectraMax iD3 Multi-Mode Microplate Reader (Molecular Devices, San Jose, CA, USA). Firefly luciferase activities were normalized by the values of Renilla luciferase.

## Bone analysis and staining

Mouse limbs and spines were collected for ex vivo μCT imaging using a vivaCT40 scanner (Scanco USA, Wayne, PA, USA). The scanned images were segmented for reconstruction to access the relative bone volume (BV/TV, %) by 3D visualization and analysis using Amira (FEI, Thermo Fisher Scientific). To perform quantitative measurements, 170 and 200 slices were used for the femur trabecular bone and L5 spine, respectively. Cortical bone thickness (Ct. Th, mm) was analyzed at a standardized location of 30 slices near the midshaft. Skeletal preparation, fixation, and embedding for paraffin sections were performed as described (*Maruyama et al., 2010*; *Yu et al., 2005a*; *Mirando et al., 2010*; *Maruyama et al., 2013a*; *Yu et al., 2010*; *Yu et al., 2005b*; *Yu et al., 2007*; *Maruyama et al., 2013b*; *Maruyama et al., 2016*; *Maruyama et al., 2017*). Samples were subject to hematoxylin and eosin staining for histology, TRAP staining, van Kossa staining, or immunological staining with avidin: biotinylated enzyme complex (*Maruyama et al., 2010*; *Yu et al., 2005a*; *Maruyama et al., 2013a*; *Yu et al., 2005b*; *Yu et al., 2007*; *Maruyama et al., 2016*; *Maruyama et al., 2017*; *Chiu et al., 2008*; *Fu and Hsu, 2013*; *Russell, 1972*; *Fu et al., 2014*). The immunological staining was visualized by enzymatic color reaction or fluorescence according to the manufacturer's specification (Vector Laboratories, Burlingame, CA, USA). After TRAP staining, N.Oc/T.Ar, N.Oc/BS, and Oc.S/BS (%) were determined for statistical significance. Rabbit antibodies Collagen I (LSL-LB-1190, Cosmo Bio Co., LTD., Japan; 1:2000), Cathepsin K (ab19027, Abcam, Cambridge, MA, USA; 1:50); mouse antibodies Osteopontin (MPIIIB10, Hybridoma Bank, IA, USA; 1:500) were used in these analyses. Images were taken using Leica DM2500 and DFC310FX imaging system (Leica, Bannockburn, IL, USA) and Zeiss Axio Observer microscope (Carl Zeiss, Thornwood, NY, USA). BFR was examined by double labeling of Alizarin Red S and Calcein Green, injected intraperitoneally with a 7 day interval. The labeled samples were embedded without decalcification for frozen sections (SECTION-LAB Co. Ltd, Japan; *Maruyama et al., 2013a*; *Maruyama et al., 2016*; *Maruyama et al., 2017*), followed by analyzed under a fluorescent microscope using an OsteoMeasure morphometry system (OsteoMetrics, Atlanta, GA, USA) to determine Mineral Apposition Rate (MAR, BFR/BS).

## Statistics and reproducibility

Microsoft Excel 2010 was used for statistical analysis. The significance was determined by two-sided student t-tests. A p-value less than 0.05 was considered statistically significant. Before performing the t-tests, the normality of the data distribution was first validated by the Shapiro-Wilk normality test. Analysis of samples by µCT was performed by a technician who is blinded to the condition. No randomization, statistical method to predetermine the sample size, and inclusion/exclusion criteria defining criteria for samples were used. At least three independent experiments were performed for statistical analyses of the animal tissues described in figure legends. Statistical data were presented as mean ± SD. The miPath v.3 KEGG Reverse Search with Search Pathway: mmu04380 and Method: TarBase v7.0 was used to identify the candidates that are both miR-27a targets (TarBase v7.0) and genes associated with osteoclast differentiation (KEGG: mmu04380).

## Materials and correspondence

Correspondence and material requests should be addressed to W.H. and T.M.

## Acknowledgements

We thank Chyuan-Sheng Lin, Keiko Kaneko, Ya-Hui Chiu, Michael Thullen, and Hsiao-Man Ivy Yu for assistance in transgenic mouse strains, plasmid DNA construction and purification, FACS, µCT scanning, and imaging analysis, respectively. Research reported in this publication is supported by the National Institute of Dental and Craniofacial Research of the National Institutes of Health under award numbers R01DE015654 and R01DE026936 to WH. and R21DE028696 to TM.

## Additional information

### Funding

| Funder | Grant reference number | Author |
|---|---|---|
| National Institutes of Health | DE015654 | Wei Hsu |
| National Institutes of Health | DE026936 | Wei Hsu |
| National Institutes of Health | DE028696 | Takamitsu Maruyama |

The funders had no role in study design, data collection and interpretation, or the decision to submit the work for publication.

### Author contributions

Shumin Wang, Conceptualization, Data curation, Formal analysis, Investigation, Writing - original draft; Eri O Maruyama, John Martinez, Trunee Hsu, Data curation, Formal analysis, Investigation; Justin Lopes, Data curation; Wencheng Wu, Software, Methodology; Wei Hsu, Takamitsu Maruyama, Conceptualization, Data curation, Supervision, Funding acquisition, Writing - original draft, Project administration, Writing - review and editing

### Author ORCIDs

Shumin Wang ⓘ http://orcid.org/0000-0003-0217-6600
John Martinez ⓘ http://orcid.org/0000-0003-0040-8519
Trunee Hsu ⓘ http://orcid.org/0000-0001-5790-1836
Wei Hsu ⓘ http://orcid.org/0000-0001-6738-6030
Takamitsu Maruyama ⓘ http://orcid.org/0000-0001-7322-8821

### Ethics

Care and use of experimental animals described in this work comply with the recommendations in the Guide for the Care and Use of Laboratory Animals of the National Institutes of Health. All of the

animals were handled according to approved institutional animal care and use committee (IACUC) protocols (#102402) of the University of Rochester and protocols (#21-005) of the Forsyth Institute.

### Decision letter and Author response
Decision letter https://doi.org/10.7554/eLife.79768.sa1
Author response https://doi.org/10.7554/eLife.79768.sa2

## Additional files

### Supplementary files
• Supplementary file 1. Primers for RT-PCR analysis of MIR23a~27a ~24–2.
• MDAR checklist

### Data availability
All data generated or analyzed during this study are included in the manuscript and supporting file.

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
