## [Editor Report]

The study has been constructively revised with compelling evidence. The addition of pit resorption and actin formation assay along with the status of expression of osteoclast fusion markers have made the article more systematic making the findings fundamentally impactful. The study is novel and contributes significantly to bone biology research.

---

## [Decision Letter]

**Decision letter after peer review:**

Thank you for submitting your article "MicroRNA-27a is essential for bone remodeling by modulating p62-mediated osteoclast signaling" for consideration by *eLife*. Your article has been reviewed by 3 peer reviewers, one of whom is a member of our Board of Reviewing Editors, and the evaluation has been overseen by Mone Zaidi as the Senior Editor. The reviewers have opted to remain anonymous.

Essential Revisions

The reviewers have discussed their reviews with one another, and the Reviewing Editor has drafted this to help you prepare a revised submission. Please address the reviewer recommendations and provide a point-to-point rebuttal.

*Reviewer #1 (Recommendations for the authors):*

1. The authors suggest miR-27a supplementation as a new therapeutic approach toward osteoporosis but authors should ensure that the supplementation or overexpression of miR27a would not alter any other physiological process and the status of expression in different tissues should also be taken into account.

2. miR27a did not downregulate the luciferase activity associated with the 3'UTR of Cathepsin K but ∆miR-23a~27a show a ~3-fold increase in the number of Cathepsin K-expressing osteoclast cells lining the trabecular bone surface. What could be the possible reason for that?

3. The authors should also indicate the underlying pits or resorption pit area in TRAP-positive cells.

*Reviewer #2 (Recommendations for the authors):*

I have a few queries and suggestions to improve the quality of the manuscript.

1. Authors mentioned that "miR-23a~27a are not required for the developmental processes" but didn't show any data for this claim. How do authors assess the effect of miR-23a~27a on skeletal development?

2. In figure 1 authors showed that "ΔmiR-23a~27a" animals showed a significant bone loss at 3 months. Do these animals start losing bone at an early stage or during the developmental stage these animals have less trabecular bone?

3. From the micro-CT data (Figure 1) it's clearly visible that female animal loses more femur trabecular bone compared to male animals. What could be the possible explanation?

4. Does miR-27a expression altered in different pathological conditions, like osteoporosis or osteopetrosis?

5. What will be the clinical advantage of miR-27a over standard antiresorptive like bisphosphonate?

*Reviewer #3 (Recommendations for the authors):*

The author should show a mechanism by which miR27a regulates p62 signaling in osteoclast cells.

---

## [Author Response]

Reviewer #1 (Recommendations for the authors):1. The authors suggest miR-27a supplementation as a new therapeutic approach toward osteoporosis but authors should ensure that the supplementation or overexpression of miR27a would not alter any other physiological process and the status of expression in different tissues should also be taken into account.

We agreed with the reviewer that supplementation or overexpression of miR27a should not alter other physiological processes in different tissues. In the Discussion section, we already described the negative effect of the overexpression of miR-23a on bone formation. We also mentioned that the high levels of miR-23a in the mast cells lead to bone loss through the release of extracellular vesicles by neoplastic mast cells. Taking into consideration of possible effects on other tissues and processes, we, therefore, also indicate that targeting miR-27a on the bone resorption surface with synthetic compounds e.g. bisphosphonates, or osteoclast-targeting molecules e.g. acid octapeptides with aspartic acid is critical for future clinical application. These statements have been incorporated in the revised Discussion section.

2. miR27a did not downregulate the luciferase activity associated with the 3'UTR of Cathepsin K but ∆miR-23a~27a show a ~3-fold increase in the number of Cathepsin K-expressing osteoclast cells lining the trabecular bone surface. What could be the possible reason for that?

The upregulation of Ctsk might be attributed to the direct regulation of mRNA degradation mediated by miR-27a. Alternatively, this increase could be the outcome of enhanced osteoclast differentiation/maturation caused by the loss of miR-27a. Our results indicated that Ctsk is not the direct target of miR-27a. Therefore, the increase of CtsK^+^ cells is most likely the consequence of enhanced osteoclastogenesis in ∆miR-27a mice.

3. The authors should also indicate the underlying pits or resorption pit area in TRAP-positive cells.

Based on the reviewer’s suggestion, we have performed the resorption pit assay to show that the resorption rate is elevated by the deletion of miR-27a. This elevation could be alleviated by the knockdown of p62 suggesting that the miR-27a-p62 axis regulates the resorption activity of osteoclast cells. The new data has been included in the revised Figure 6E.

Reviewer #2 (Recommendations for the authors):I have a few queries and suggestions to improve the quality of the manuscript.1. Authors mentioned that "miR-23a~27a are not required for the developmental processes" but didn't show any data for this claim. How do authors assess the effect of miR-23a~27a on skeletal development?

As suggested by the reviewer, we have included the data examining the skeletal development of ∆miR-23a~27a mice. Skeletal staining at P0 showed no noticeable structural difference between the WT and ∆miR-23a~27a mice. The skull width and length as well as, the humerus, radius, femur, and tibia appear comparable in WT and ∆miR-23a~27a. The results indicating miR-23a and miR-27a dispensable during skeletal development have been presented as the new Figure S1.

2. In figure 1 authors showed that "ΔmiR-23a~27a" animals showed a significant bone loss at 3 months. Do these animals start losing bone at an early stage or during the developmental stage these animals have less trabecular bone?

Based on the reviewer’s comment, we performed additional μCT analysis for the newborn WT and ∆miR-27a femur. The results showing the mutant has comparable bone volume compared to the control have been included in the new Figure S1E.

3. From the micro-CT data (Figure 1) it's clearly visible that female animal loses more femur trabecular bone compared to male animals. What could be the possible explanation?

Based on the statistical evaluation, the bone volume is ~46% in miR-23a~27a females and ~49% in miR-23a~27a males compared to the controls. The 3D rendering images could be deceiving sometimes. There is no statistically significant difference in gender-associated severity of the bone loss.

4. Does miR-27a expression altered in different pathological conditions, like osteoporosis or osteopetrosis?

The reduced expression of miR-23a and miR-27a in osteoporosis patients has been described in the Introduction and Discussion. According to the comment, we have further the literature search to find miR-23a as one of the miRNAs that is increased significantly in osteopetrosis patients (Ou, Eur. J. Hum. Genet., 2014). This notion implies the physiological importance of the miR-23a cluster to regulate bone remodeling in health and disease. We have included additional descriptions in the revised Discussion.

5. What will be the clinical advantage of miR-27a over standard antiresorptive like bisphosphonate?

Bisphosphonates are widely used as an effective treatment for osteoporosis (Khosla, Lancet, 2017). However, there are rare side effects, including atypical femur fractures and osteonecrosis of the jaw. In addition, the efficacy after 5 years of treatment remains uncertain (Izano, JAMA network open, 2020). A better understanding of the bone remodeling process could help maximize anti-fracture efficacy and minimize adverse skeletal effects. The current study has demonstrated that miR-27a is essential for the miRNA-mediated regulation of osteoclast differentiation and maturation. The loss of miR-27a alone is sufficient to cause osteoporotic bone loss. Because miR-27a downregulation is associated with postmenopausal women, supplementation or osteoclast-specific delivery of miR-27a represents an attractive approach for targeting postmenopausal osteoporosis in precision medicine. Therefore, we have elaborated on these points in the revised Discussion.

Reviewer #3 (Recommendations for the authors):The author should show a mechanism by which miR27a regulates p62 signaling in osteoclast cells.

Based on the reviewer's comment, we have performed qRT-PCR for Rank, Dcstamp, Ctsk, and Calcr to further decipher the mechanistic regulation mediated by the miR-27a-p62 axis. The miR-27a deletion elevated the expression of Rank, but this elevation is reduced by p62 knockdown, suggesting a role of the miR-27a-P62 axis in regulating the RANKL signaling sensitivity in osteoclast cells. Enhanced expression of Ctsk and Calcr in ΔmiR-27a osteoclast cells could be alleviated by the knockdown of p62, implying that miR-27a regulates osteoclast maturation through modulation of p62. However, the enhanced expression of Dcstamp in ΔmiR-27a OC cells was not affected by the p62 knockdown, indicating the miR-27a-mediated regulation of cell fusion is independent of p62. These new data have been incorporated into Figure 6 of the revised manuscript.